# Comparative Analysis of the Immune Response and the Clinical Allergic Reaction to Papain-like Cysteine Proteases from Fig, Kiwifruit, Papaya, Pineapple and Mites in an Italian Population

**DOI:** 10.3390/foods12152852

**Published:** 2023-07-27

**Authors:** Ivana Giangrieco, Maria Antonietta Ciardiello, Maurizio Tamburrini, Lisa Tuppo, Chiara Rafaiani, Adriano Mari, Claudia Alessandri

**Affiliations:** 1Institute of Biosciences and BioResources (IBBR), National Research Council of Italy (CNR), 80131 Naples, Italy; ivana.giangrieco@ibbr.cnr.it (I.G.); maurizio.tamburrini@ibbr.cnr.it (M.T.); lisa.tuppo@ibbr.cnr.it (L.T.); 2Associated Centers for Molecular Allergology (CAAM), 00100 Rome, Italy; pedrigna@gmail.com (C.R.); adriano.mari@caam-allergy.com (A.M.); claudia.alessandri@caam-allergy.com (C.A.); 3Allergy Data Laboratories (ADL), 04100 Latina, Italy

**Keywords:** papain-like cysteine protease, allergy, safety, protein primary structure, food industry, Act d 1, Ana c 2, Cari p 2, Cari p Papain, Fic c Ficin

## Abstract

Several plant papain-like cysteine proteases are exploited by the food, cosmetic, pharmaceutical and textile industries. However, some of these enzymes can cause allergic reactions. In this context, we investigated the frequency of sensitization and allergic reactions to some fruit and/or latex cysteine proteases, which are used as additives by the food industry to improve and modify the quality of their products. The FABER test was used to analyse the patients‘ sensitization towards five plants and, for comparison, two homologous mite cysteine proteases. In an Italian population of 341 allergic patients, 133 (39%) had IgE specific for at least one of the seven cysteine proteases under investigation. Most of the patients were IgE positive for Der p 1 and/or Der f 1 (96.38%) reported a clinical history suggestive of respiratory allergy to mites, whereas none of the subjects sensitized to the homologs from papaya, pineapple and fig reported allergy symptoms following ingestion of these foods. Only one patient referred symptoms from ingesting kiwifruit. Therefore, the obtained results showed that sensitization to the fruit enzymes was only rarely concomitant with allergic reactions. These observations, together with the literature reports, suggest that the allergy to plant papain-like cysteine proteases might mainly be an occupational disease.

## 1. Introduction

Papain-like cysteine proteases (PLCPs) are widespread in nature, having been found in plants, animals, viruses, bacteria, yeasts and protozoa. These enzymes have key physiological functions, including involvement in the regulation of protein turnover in the acidic conditions of lysosomes [1]. PLCPs are also considered of great biotechnological and commercial value due to their strong proteolytic activity against a broad range of protein substrates. Enzyme hydrolysis is often preferred over chemical methods to hydrolyse animal or plant food proteins for different reasons. For instance, to improve nutritional characteristics, to delay deterioration, to modify different functional properties (solubility, foaming, coagulation, and emulsifying capacities), to prevent undesired interactions and to change flavours and odours [2,3].

The PLCP protein family includes some well-known proteases, such as actinidin from kiwifruit, papain and chymopapain from papaya fruit and latex, bromelain from pineapple and ficin from fig fruit and latex. In nature, similar to other proteases, to prevent unwanted digestion, they are synthesized as inactive proenzymes containing a N-terminal signal peptide, followed by a propeptide region and a catalytic domain [4]. The propeptide plays important roles as an inhibitor of enzymatic activity and assisting protein folding. Removal of the propeptide is necessary to obtain the mature and catalytically active protease. When isolated from the natural source, cysteine proteases have been found as mature proteins, as reported for enzymes such as papain [5], bromelain [6,7] and actinidin [8,9]. The literature also reports [10] that, at least in some cases, the protease precursor is accumulated and stored; then, the wounding can represent an inducer of the protein maturation by removal of the propart. Details on the signal peptide and propart region can be obtained by the analysis of gene sequences.

Papain, named Cari p Papain as an allergen (www.allergome.org, accessed on 26 May 2023) [11] is the best-known cysteine protease, which was isolated as early as 1879 from the fruits of *Carica papaya* and is also the first protease for which a crystallographic structure was determined [12]. It is stabilized by three disulfide bridges and the chain is folded into two domains, one predominantly helical and the other containing a barrel, with the active site in a groove between the domains. All PLCPs are thiol-dependent peptidases. The mechanism of action involves a generally conserved catalytic Cys-His-Asn triad lying at the surface of the cleft between the two domains of the molecule [4]. In papain, this Asn residue, which is in position 175, does not seem to play an essential catalytic role for the protease activity [13]. Nevertheless, it seems to play an important role in the orientation of the imidazolium ring of the histidine in the catalytic cleft and the strict conservation of this residue might result from a combination of functional and structural constraints. Although the active site conformation is very similar in the PLCPs family, evidence is available [14] suggesting that minor structural changes can affect the substrate specificity. In general, it is known that PLCPs hydrolyze proteins with broad specificity for peptide bonds but they show a preference for an amino acid bearing a large hydrophobic side chain at the P2 position [15]. These enzymes do not accept Val in P1′.

Plant PLCPs are widely used in the food industry, pharmaceuticals, and cosmetics. For instance, the baking industry exploits these proteases because dough may be prepared more quickly if the gluten has been partially hydrolyzed [16]. In the textile industry, papain can be used for processing wool, boiling off cocoons and refining silks [17]. Bromelain has been exploited in many applications in the food, beverage, tenderization, cosmetic, pharmaceutical and textile industries [18]. Actinidin can be used to tenderize meat and also improve emulsion stability, texture, and organoleptic properties of this food [19,20]. Ficin can have an application in the production of some traditional cheeses and milk protein hydrolysates for special food preparations [21].

Several cysteine proteases of the PLCP family have been reported as a cause of allergic reactions, including the group of mite allergens, such as the major dust-mite allergen Der p 1 [22] and food allergens such as actinidin, bromelain, ficin, and papain [23,24,25]. Some of them are considered important allergens and have been registered by the WHO/IUIS Allergen Nomenclature Subcommittee (http://allergen.org/, accessed on 26 May 2023) [26], which assigned them the allergen name. This list includes Der p 1 and Der f 1 [27], isolated from the mites *Dermatophagoides pteronyssinus* and *Dermatophagoides farinae*, respectively. The list also includes some plant homologous enzymes, such as actinidin, bromelain, and chymopapain, isolated from kiwifruit (*Actinidia deliciosa*), from the fruit and stem of the pineapple (*Ananas comosus*), and from the fruit of papaya (*Carica papaya*), which were registered by the WHO/IUIS with the allergen names Act d 1 [28], Ana c 2 [29], and Cari p 2 [30], respectively. Some additional components of this protein family were registered by the WHO/IUIS as allergens, but they are not included in this study. A number of PLCPs have been reported to cause allergic reactions, although they are currently not registered in the official list of the WHO/IUIS. For instance, this is the case of papain [11] from papaya and ficin (www.allergome.org, accessed on 26 May 2023) from fig (*Ficus carica*), for which we have here used the names Cari p Papain and Fic c Ficin.

PLCPs can cause occupational allergic reactions due to exposure to airborne particles, as reported, for instance, for papain [11,31] or bromelain [32]. The same allergens can cause nonoccupational reactions due to the ingestion of foods containing the allergenic protein [33]. Sometimes, allergenic proteins are contained in foods which do not represent their natural source because they are added to foodstuffs for biotechnological purposes and the consumer could not be aware of their presence. In this case, the PLCP represents a hidden allergen which might be a food-safety problem for sensitized consumers. Despite their uncommonness, the isolation and characterization of the IgE binding capacity of all the proteolytic enzymes from allergenic sources remain an incomplete and challenging task [34].

In this context, the aim of this study was the investigation of the frequency of sensitization by the measurement of IgE-positive results, and of allergic reactions to PLCPs that are present in raw plant foods and/or in products from the food industry. Therefore, this is an important subject of investigation because it falls into the food-safety topic. Here, the frequency of sensitization to five plant PLCPs (Act d 1, Ana c 2, Cari p Papain, Cari p 2, and Fic c Ficin) and, for comparison, two mite homologous enzymes, Der p 1 and Der f 1, was analysed by exploiting the FABER test, which allows for the detection of a specific IgE in the patients’ sera. A random Italian population reporting allergic symptoms was selected for this investigation. The study includes a comparative analysis of some structural features of the above-listed PLCP proteins. The results obtained in the analysed population show that compared to the mite homologs, the frequency of sensitization to the plant food proteases is lower and the clinical data suggest that the allergic reactions to the ingestion of plant food PLCPs might be rare.

## 2. Materials and Methods

The PLCPs analysed in this study are from *D. pteronyssinus* and *D. farinae* (Der p 1 and Der f 1, Indoor Biotechnologies, Cardiff, UK), *C. papaya*, (Cari p Papain and Cari p 2, Sigma-Aldrich, Milan, Italy), *A. comosus*, (Ana c 2, Sigma-Aldrich), and *F. carica* (Fic c Ficin, Sigma-Aldrich). Act d 1 was purified from *A. deliciosa*, following an already reported procedure [9].

### 2.1. Analysis of Papain-like Cysteine Proteases Primary Structure

PLCPs are encoded by multigenic families expressing several protein isoforms. A multiple sequence alignment of all the available isoforms of cysteine proteases deriving from each organism under investigation would be very complex. Therefore, to make the analysis easier, a representative sequence for each organism/type of enzyme has been chosen and used for comparative investigation. As a general criterion of isoforms selection, curated sequences of the molecules really detected at the protein level have been chosen. Protein isoforms were registered in UniProt with a code having P as the first letter, followed by a number, where preferred, when available. In fact, this kind of code is generally a synonym of a reviewed protein sequence corresponding to a molecule isolated as a protein (rather than as a DNA sequence) and, very likely, it was present in high amounts in the natural source. Quite recently, the coding system of UniProt has been changed and, therefore, it is now more difficult to deduce some information from the accession number format. For instance, simply looking at the protein code, it is now not possible to understand if the sequence is gene-deduced or if the natural protein was isolated and characterized.

Amino acid sequences representative of PLCPs of mites, kiwifruit, papaya, pineapple and fig were selected and aligned choosing “Align” on the menu of the UniProt website (www.uniprot.org, accessed on 26 May 2023). Thereby, a multiple sequence alignment was obtained with the program Clustal O (Clustal Omega). The output data also contained additional information, including the percentage of amino acid identity between the different protein sequences.

### 2.2. Specific IgE Detection with the FABER^®^ Multiplex Testing System

The FABER test (Allergy Data Laboratories, ADL, S.r.l., Latina, Italy) is a multiplex in vitro serological test allowing for the detection of specific IgE antibodies produced by allergic subjects [35,36]. The FABER biochip is used for an ELISA-like testing procedure allowing for the detection of a specific IgE in each of the 244 allergenic preparations, in addition to some experimental allergens, in a single run [37,38]. IgE detection is performed by using an alkaline phosphatase-conjugated anti-IgE. The final read out for each allergen is obtained with the use of an optical scanner and elaborated with Raptor v1 software. The detected signals are interpolated with values obtained with an internal standard reference curve of IgE present in each biochip. Signals intensities are then used to calculate the arbitrary units, FIU (FABER International Units), correlated with the level of IgE bound to each allergen spot.

The data here described were obtained from a set of biochips including, in addition to the standard 244 allergenic preparations, also allergens spotted for experimental purposes, such as the ficin protein (Fic c Ficin). Before the immobilization on the FABER biochip, ficin and the other proteins tested for experimental purposes were coupled to nanobeads following the same procedure applied to all the allergenic preparations.

### 2.3. Patients

The IgE profile of 341 Italian patients tested with the FABER system was analysed. Their average age was 33.5 years and 186 (54.5%) patients were female. This was considered a random cohort, representing a general population of allergic subjects since no selection criteria were applied. A reliable clinical history was available for each one. The symptoms reported by patients, and other details including the use of medications, were recorded in a specific questionnaire, as described in a previous article [39]. All investigations were carried out according to the World Medical Association Declaration of Helsinki about Ethical Principles for Medical Research Involving Human Subjects. All patients gave their informed consent to the use of their clinical data in an anonymous form. They also gave informed consent to the use of their anonymized leftover serum for research purposes. Since the study was carried out within a routine activity, formal approval by an external ethical committee was not required.

## 3. Results

Table 1 shows some details of the plant and mite PLCPs analysed in this study.

### 3.1. Comparison of Primary Structures

Figure 1 shows the alignment of two cysteine proteases from mites and five homologous enzymes from plant tissues. The sequences of mature proteins, where signal and propart have been deleted, were used for this multiple alignment. In general, the alignment shows that the length of the mature molecules is poorly conserved, ranging from 212 (Cari p papain) to 254 (Act d 1) residues and several gaps/insertions are required to align the sequences. In addition, most of the residues are not conserved in all the aligned sequences. In fact, a few amino acids, including the catalytic cysteine (C) and histidine (H) dyad, which are highlighted in yellow in Figure 1, are conserved. The Asn residue (N), which is suggested to play an important role in the orientation of the substrate during catalysis [13], is conserved in the aligned sequences, except in the Ana c 2 isoform, which has a Lys (K) in that position (Figure 1).

Actually, the analysis of 14 sequences of Ana c 2 reveals the level of differences between the individual isoforms and suggests possible immunological and functional variability. The comparison shows that the only isoform lacking the Asn residue involved in the catalytic activity is the one with accession number P14518 (Appendix A). The Ana c 2 isoform F1KD58 is truncated and, therefore, by lacking that sequence region, we cannot evaluate the presence of the Asn residue. Therefore, out of 14 Ana c 2 sequences, 12 show the Asn residue involved in the catalytic activity and generally conserved in papain-like cysteine proteases. The alignment in Appendix A also reveals that three isoforms, out of 14, contain the amino acid motif required for N-linked glycosylation to occur (N-X-S/T). This structural determinant can affect the in vitro IgE tests, thus giving positive results, but they do not have a clinical meaning [67].

The Ana c 2 isoform registered by WHO/IUIS as an allergen (Ana c 2.0101) is the molecule with UniProt accession number O23791, which is devoid of the N-glycosylation site found in the isoforms with accession numbers P14518, O23799, and O81084 (Appendix A). The amino acid sequence identity (%) between the isoforms of Ana c 2 aligned in Appendix A ranges from 100 to 68% (Appendix A). Der p 1 and Der f 1 also contain a N-glycosylation site that is shifted towards the N-terminal region, with respect to the plant homologous enzymes (Figure 1).

Table 2 shows the amino acid sequence identity (%) between the seven papain-like proteases analysed in this study. A high value, corresponding to 82.43%, is observed only when the two mite allergens are compared. Lower values, ranging from 41.15% to 59.43%, are observed when the plant enzymes are compared. Very low values, ranging from 29.06% to 38.35%, are registered when plant proteases are compared with the mite allergens.

### 3.2. Allergy Symptoms of the Analysed Population

The allergic symptoms referred to by patients on the occasion of the first visit, during the collection of the anamnesis, are summarized in Figure 2. This figure shows that about 80% of the analysed population (341 patients) suffered from rhinitis. Their medical history seemed to suggest a respiratory allergy to house dust-mite (HDM) allergens due to a constantly stuffy nose, sneezing in the morning upon waking up, sniffing, and nasal itching.

Symptoms like conjunctivitis, asthma, and urticaria were also reported with a frequency of around 30%. Rhinitis and conjunctivitis were mostly associated with patients complaining of symptoms in the spring. Lower frequency values were observed for symptoms like oral allergy syndrome (OAS), gastritis, enteritis, eczema, colitis, angioedema, and anaphylaxis due to other suspected allergenic sources where PLCP has not been reported yet. Kiwifruit ingestion caused OAS in one patient and OAS and urticaria in another one. One patient referred to fig pruning itch and two patients claimed OAS ingesting fig fruit. Twenty subjects (5.8%) referred to not eating fruits, including fig, kiwi, papaya, and pineapple. Among the 321 remaining patients, none claimed symptoms of ingesting papaya or pineapple.

### 3.3. Results of Specific IgE Detection by In Vitro Tests

The sera (341) from patients reporting allergic symptoms towards any food or inhalant source were analysed with the FABER test [35,37]. The analysis revealed that 133 of them (39%) were positive for at least one of the seven cysteine proteases under investigation, namely Act d 1, Ana c 2, Cari p 2, Cari p Papain, Der f 1, Der p 1, and Fic c Ficin (Appendix A). Table 3 highlights that the allergens most frequently recognized by specific IgE are those from mites, and Der f 1 and Der p 1, which were IgE positive in 73 (21.4%) and 67 (19.6%) patients. In addition, 58 patients (17%) were cosensitized to both the mite cysteine proteases. The kiwi and papaya proteases, Act d 1, Cari p 2, and Cari p Papain, were recognized by specific IgE with a low frequency, corresponding to about 0.9%, 2.6%, and 1.5%, respectively. The sensitization to the protease Ana c 2, considered a cross-reactive carbohydrate determinant (CCD) marker, was detected in nine sera (12%).

Despite sensitization to the cysteine proteases, no patients reported allergy symptoms to pineapple, papaya, and fig, whereas the three subjects reporting OAS against fig (patients n. 77 and 252, in Appendix A) and reactions at pruning (patient n. 153, in Appendix A) did not have specific IgE for Fic c Ficin. Out of three patients with IgE specific for Act d 1, only one (patient n. 3, in Appendix A) reported allergy OAS and urticaria following kiwifruit ingestion. Appendix A also shows that out of 83 patients with IgE specific for Der p 1 and/or Der f 1, 80 (96.38%) reported allergy symptoms towards mites.

## 4. Discussion

A comparative analysis of the primary structure of seven allergenic PLCPs investigated in this study shows a low protein sequence conservation among them, except a quite high identity between the two mite allergens, Der p 1 and Der f 1. Obviously, the catalytic residues are conserved in the same position in the investigated proteases. In line with these structural findings, the immunological behaviour of the two mite cysteine proteases is almost overlapping. In contrast, the plant homologous proteins show individual immunological behaviours, when compared with each other and with the mite Der p 1 and Der f 1. This is in line with the observation that a high conservation of antigenic epitopes, associated with similar IgE-binding results, is frequently found in proteins with high sequence identities.

Some components of the PLCP family have been reported to be glycosylated. The literature reports [67] that these oligosaccharides, known as CCD, do not have a clinical meaning; that is, the patients producing IgE specific for these carbohydrates do not show allergic reactions when they are exposed to the proteins recognized by these antibodies. Nevertheless, this post-translational modification can affect the results of IgE tests and interfere with the diagnostic conclusions. In other words, we can observe a positive IgE result on specific glycosylated allergenic proteins, which is not correlated with allergic symptoms. Therefore, the presence of CCD in an allergenic protein can generate diagnostic confusion. In the case of some PLCPs, such as bromelain (Ana c 2), the confusion can be even higher because we have observed that only a fraction of the isoforms of this protease has a N-glycosylation site, which can bear a carbohydrate and, therefore, can contribute to generating molecular heterogeneity.

It is worthy of note that Ana c 2 is generally considered a CCD marker [68,69], exploited to reveal the presence in the patient serum of IgE specific for CCD. However, the analysis of the primary structure of different isoforms of this protein has revealed that most of them do not have N-glycosylation sites. The only isoform officially recognized as an allergen by WHO/IUIS is Ana c 2.0101 (UniProt accession number O23791) and the mature form of this protein does not have the N-glycosylation site found in other isoforms. It only has a N-glycosylation site at the end of the propart sequence reported in UniProt but this protein region is removed in the mature protein. The mite cysteine proteases, Der p 1 and Der f 1, also show a N-glycosylation site that can bear carbohydrates chains. The literature reports that allergens, like Der p 1, are glycosylated and the carbohydrate moieties could play a role in their recognition by innate immune cells [40]. The presence of different types of short carbohydrates at the N-glycosylation site in Der f 1 has also been suggested [46]. However, the carbohydrates bound to the mite cysteine proteases should be different compared to those of plant proteins and the data available do not suggest that they are recognized by CCD-specific IgE. Anyway, although CCDs, and the IgE recognizing them, do not seem to have clinical relevance, an increase in the knowledge on this topic can be helpful for the correct interpretation of the immunologic tests and can be useful in clinical practice.

The analysis of a random population of 341 Italian patients complaining of allergic reactions towards any food or inhalant source revealed that rhinitis was the most reported symptom (about 79%), followed by conjunctivitis and urticaria. In this context, FABER is the only diagnostic test that has been exploited for a comparative analysis of specific IgE towards plant fruit PLCPs from kiwifruit, papaya, pineapple, and fig, as well as the homologs Der p 1 and Der f 1 from mites. Out of 341 patients, 39% had IgE specific for at least one of the seven cysteine proteases under investigation, thus suggesting that the sensitization towards this protein family is not of negligible meaning. However, the results obtained for the individual allergens highlighted that the two mite homologs displayed the highest frequency of IgE-positive results (around 20%), with the patients sensitized to at least one of the two mite proteases corresponding to 24.3%. In line with the high sequence identity between Der p 1 and Der f 1, most of the patients who were IgE positive to a mite protease were cosensitized to the other one. Such findings are well known since their discovery [47,70]. It is worthy of note that the IgE results obtained for the mite proteases, combined with clinical data, reveal a high positive predictive value of Der p 1 and Der f 1 for clinical relevancy. This observation suggests that these two allergens, to which the exposure route is generally inhalation [44], provide an important contribution to the prevalence of symptoms common to mite allergy, such as rhinitis, bronchial asthma, and conjunctivitis.

In contrast to Der p 1 and Der f 1, the food PLCPs did not show the same high correlation between IgE positivity and allergy reactions. For instance, the frequencies of IgE-positive results obtained for the two cysteine proteases from papaya fruit, Cari p 2, and Cari p Papain, and for the pineapple protein, Ana c 2, were 2.6%, 1.5%, and 12%, respectively, but none of the patients reported symptoms following the exposure to these foods. We cannot exclude that the higher value (12%) of IgE-positive results obtained for Ana c 2 could be affected by CCD-specific IgE that could be contained in some patients’ sera. Nevertheless, we observe IgE positivity not correlated with clinical allergy not only on Ana c 2 but also on Cari p 2, Cari p Papain, and Fic c Ficin.

The frequency of IgE-positive results obtained by testing Act d 1, was lower (0.88%) compared to the homologs. However, only one of these patients reported allergy symptoms (OAS and urticaria) after the kiwifruit consumption. A second patient reported OAS after kiwifruit ingestion but the IgE negative tests to Act d 1 suggest that the allergy symptoms have to be attributed to other possible allergens contained in this food. In the case of Act d 1, we observe that the number of patients IgE positive to the allergen, and clinically allergic to kiwifruit, is low and does not allow for the drawing of meaningful conclusions.

A non-negligible percentage of patients (8.5%) was IgE positive to the fig cysteine protease but none of them reported allergy symptoms following the exposure to this fruit. Actually, out of 341, three patients (0.88%) claimed allergy reaction to fig, two of them reporting OAS to the fruit ingestion and the third one reported an itchy reaction following skin contact to pruning but none of them was IgE positive to Fic c Ficin. Indeed, the literature reports describe that cysteine proteases can evoke itch in the absence of any allergic reaction [71,72]. Therefore, at least in the investigated population of allergics, and on the basis of the in vitro tests here reported, the fig cysteine protease does not appear to be the culprit of the reported allergy reactions due to fig exposure.

In summary, as reported for other allergens [38,73], a structural corecognition between fig, papaya, and pineapple cysteine proteases and homologs from other sources, which is not necessarily associated with allergy reactions, is conceivable to occur in some cases. Indeed, the comparison of the primary structures of these food proteases shows a very low sequence identity, suggesting low structural similarity at the primary structure level. Nevertheless, we cannot exclude similarities between the different cysteine proteases 3D structures, sufficient to induce IgE binding but insufficient to generate allergic reactions. We cannot exclude that the analysis of a larger and/or different population, and an indepth investigation of subjects who do not eat fruits, can reveal patients with allergy reactions to these plant food PLCPs. At any rate, the obtained results suggest that the prevalence of subjects with allergy reactions to these proteins is most likely low, or even null, at least in this Italian population. However, a limitation of this study is that the allergy diagnosis is based on the history provided by the patient and it was not confirmed by oral challenges and/or basophil activation assays. For this reason, a further investigation will be addressed in a near future study.

In conclusion, at least in the investigated Italian population, sensitization to the mite cysteine proteases, Der p 1 and Der f 1 occurs with a higher frequency, compared with homologs from plant foods, such as kiwifruit, pineapple, papaya and fig. More importantly, the observed correlation between clinical allergy to mites and sensitization to Der p 1 and Der f 1 is very high, thus producing a high predictive value of allergy reactions by in vitro IgE tests. In contrast, the IgE-positive results obtained for the analysed plant food homologs do not correlate with allergy symptoms towards the protease source. In fact, despite the frequent exposure to PLCPs contained in raw fruits or included as additives in processed foods, the allergic reactions were generally absent in the sensitized patients analysed in this study. The only exception was represented by a patient with allergy reactions to kiwifruit and showing positive IgE binding to Act d 1. This is in line with the literature reports describing the reactions to some fruit cysteine proteases as allergy symptoms generally associated with occupational work. Therefore, as some of these cysteine proteases are known to cause allergy reactions in the exposed workers and to be able to sensitize the atopic subjects, it is likely that they might represent a risk of allergy in some populations. Nevertheless, this occurrence was not observed in the population examined in this study. Further investigations also performed with the inclusion of additional PLCPs on the biochip of diagnostic systems will make human biomonitoring studies possible.

## Figures and Tables

**Figure 1 foods-12-02852-f001:**
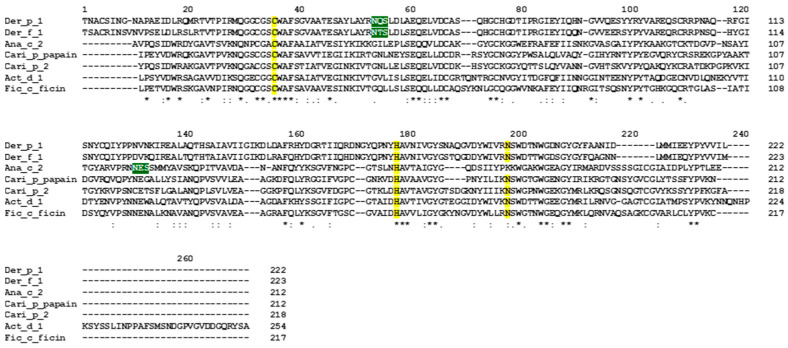
Multiple sequence alignment of representative components of PLCP allergens (mature proteins) analysed in this study. The accession numbers of the isoforms of Der p 1, Der f 1, Ana c 2, Cari p Papain, Cari p 2, Act d 1, and Fic c Ficin A, selected for this alignment, are P08176, P16311, P14518, P00784, P14518, P00785, and A0A182DW06, respectively. Three residues involved in the catalytic activity are highlighted in yellow. The N-glycosylation sites found in the aligned isoform of Ana c 2 (NES), and in the sequence of the mature form of Der p 1 (NQS) and Der f 1 (NTS), are shown in white on a dark green background. The asterisks, colons, and dots indicate the identical amino acid residues, conserved substitutions, and semiconserved substitutions. Numbers above the sequences indicate the positions. The number of residues in each sequence is indicated on the right. The order of the proteins in the alignment was fixed by the used algorithm, that is Clustal O, on the UniProt website.

**Figure 2 foods-12-02852-f002:**
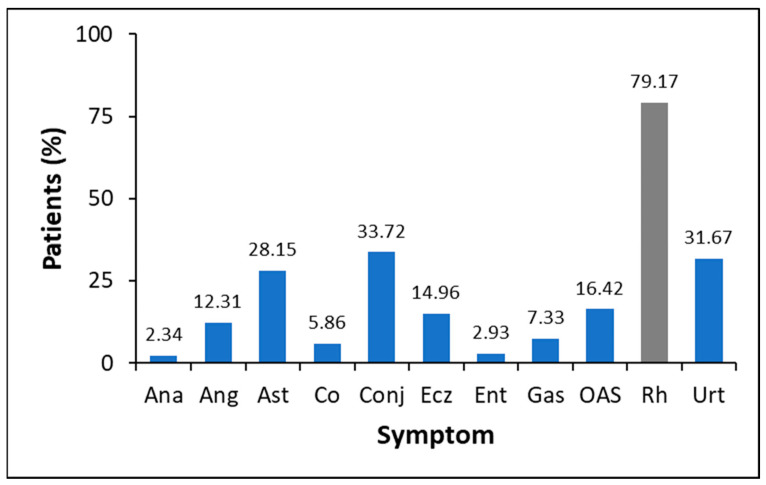
Percentage of subjects reporting allergy symptoms in the analysed Italian population of 341 patients. The symptoms referred to in the clinical history are anaphylaxis (Ana), angioedema (Ang), asthma (Ast), colitis (Co), conjunctivitis (Conj), eczema (Ecz), enteritis (Ent), gastritis (Gas), oral allergy syndrome (OAS), rhinitis (Rh), and urticaria (Urt).

**Table 1 foods-12-02852-t001:** Some details of the seven PLCPs reported as allergens and analysed in this study are summarized (the shown bibliography is representative and not exhaustive).

Protein Name	N-glycosylation Sites ^a^	Allergen Name	Organism (Scientific Name)	Source Tissue	Route of Exposure	Who Is Exposed	Reported Symptoms ^b^ Include
Der p 1	Yes, one [40]	Der p 1	*Dermatophagoides pteronyssinus*	Whole body [41]	Inhalation [42] Ingestion [43]	General population [41]	Respiratory symptoms, asthma breathlessness, angioedema, wheezing, rhinorrhea, anaphylaxis [43,44,45]
Der f 1	Yes, one [46]	Der f 1	*Dermatophagoides farinae*	Whole body [47]	Inhalation, [48], Ingestion [43]	General population [49]	Respiratory symptoms, asthma breathlessness, angioedema, wheezing, rhinorrhea, anaphylaxis [43,44,50]
Papain	Not found in the analysed isoforms	Cari p Papain	*Carica papaya*	latex, unripe fruit [51,52]	Inhalation, skin contact, [32,53] Ingestion [33,54]	General population, occupational [32]	Respiratory symptoms, rhinoconjunctivitis, asthma anaphylaxis [30,31,55]
Chymopapain	Not found in the analysed isoforms	Cari p 2	*Carica papaya*	Latex, unripe fruit [52]	Inhalation, ingestion [28,56]	General population, occupational [56]	Respiratory, gastrointestinal, anaphylaxis [28,57]
Bromelain	Yes, one (some isoforms only) [58]	Ana c 2	*Ananas comosus*	Fruit, stem [59]	Ingestion, inhalation [27]	General population, occupational [27]	gastrointestinal symptoms, asthma [60]
Actinidin	Not found in the analysed isoforms	Act d 1	*Actinidia deliciosa*	fruit [61]	Ingestion [62]	General population [63]	OAS, respiratory and gastrointestinal symptoms, anaphylaxis [8,26]
Ficin	Not found in the analysed isoforms	Fic c Ficin	*Ficus carica*	Latex, fruit (highest amount in unripe fruit [64,65]	Ingestion, occupational	General population [66], occupational	

^a^ The information refers to the mature protein. ^b^ Sometimes the symptoms towards the allergen source only were reported.

**Table 2 foods-12-02852-t002:** Amino acid sequence identity (%) between the analysed PLCPs. The identity values were calculated with the Clustal O algorithm on the UniProt website, using the amino acid sequence of mature proteins aligned in Figure 1.

	Der p 1	Der f 1	Ana c 2	Cari p Papain	Cari p 2	Act d 1	Fic c Ficin
**Der p 1**	100						
**Der f 1**	82.43	100					
**Ana c 2**	29.06	30.05	100				
**Cari p Papain**	29.21	30.20	41.15	100			
**Cari p 2**	32.21	32.69	51.43	59.43	100		
**Act d 1**	34.13	36.06	47.14	49.29	51.15	100	
**Fic c Ficin**	38.35	37.86	50.72	50.95	51.87	56.94	100

**Table 3 foods-12-02852-t003:** Number of patients IgE positive to the seven analysed proteases. Data are from a population of 341 allergics tested with the FABER diagnostic system.

	Der p 1	Der f 1	Ana c 2	Cari p Papain	Cari p 2	Act d 1	Fic c Ficin
**Der p 1**	67						
**Der f 1**	58	73					
**Ana c 2**	10	10	41				
**Cari p Papain**	2	2	3	5			
**Cari p 2**	3	2	5	2	9		
**Act d 1**	0	0	3	1	0	3	
**Fic c Ficin**	6	6	4	0	0	0	29

## Data Availability

Data are contained within the article or Appendix A.

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
