# Peer review of "Comparative Analysis of the Immune Response and the Clinical Allergic Reaction to Papain-like Cysteine Proteases from Fig, Kiwifruit, Papaya, Pineapple and Mites in an Italian Population"

_foods, 2023, doi:10.3390/foods12152852_

Round 1
Reviewer 1 Report
The paper entitled ‘Comparative analysis of the immune response and the clinical allergic reaction to papain-like cysteine proteases from fig, kiwifruit, papaya, pineapple and mites in an Italian population’ reported the frequency of sensitization and allergic reactions to some fruit and/or latex cysteine proteases. This work in interesting. I have no specific concern but it is mandatory to correct the manuscript in this point:
1. Please add the structural of the target protein.
2. In Fig. 2, please remove the background horizontal line and label the numbers.
Author Response
We thank very much the Reviewer for the revision of our manuscript and for all the comments and suggestions, which have been carefully considered and exploited to improve the manuscript. We hope that this revised version is now suitable for publication.
Please find below our detailed answers to the Reviewer’s comments.
Reviewer 1
Comments and Suggestions for Authors
The paper entitled ‘Comparative analysis of the immune response and the clinical allergic reaction to papain-like cysteine proteases from fig, kiwifruit, papaya, pineapple and mites in an Italian population’ reported the frequency of sensitization and allergic reactions to some fruit and/or latex cysteine proteases. This work in interesting. I have no specific concern but it is mandatory to correct the manuscript in this point:
COMMENT 1
- Please add the structural of the target protein.
ANSWER 1
The following text has been added in the Introduction section at lines 66-70 “Although the active site conformation is very similar in the PLCPs family, evidence is available [14] suggesting that minor structural changes can affect the substrate specificity. In general, it is known that PLCPs hydrolyze proteins with broad specificity for peptide bonds, but they show preference for an amino acid bearing a large hydrophobic side chain at the P2 position [15]. These enzymes do not accept Val in P1′.”
COMMENT 2
- In Fig. 2, please remove the background horizontal line and label the numbers.
ANSWER 2
In Figure 2, the horizontal lines were deleted and the columns have been labeled with numbers.
Reviewer 2 Report
This manuscript describes results obtained with a cohort of Italian patients regarding sensitization to cysteine protease allergens. The cysteine proteases from dust mite, Der p 1 and Der f 1, were shown to be more clinically significant than homologous cysteine proteases from several fruits – kiwi, pineapple, papaya, and fig. The manuscript was quite interesting to read and the observations were noteworthy in the sense that the investigation indicated that proteins with similar biological functions can have differing sequence identities and differing clinical significance. This manuscript could use some improvement.
(1) The authors do not describe the limitations of their study. The major limitation is that the basis for the allergy diagnosis is the history provided by the patient. The findings would be more impactful if oral challenges had been conducted to confirm the allergy or if basophil activation assays had been performed using the sera of patients.
(2) While the manuscript is interesting to read, the authors seem to repeat key observations several times within the text. The manuscript could be more succinctly presented without losing any major points.
(3) Table 1 – Papain is known to be an ingestion allergens – admittedly on a rare basis. Should be included in this table. Key citations are Mansfield and Bowers, J Allergy Clin Immunol 71:371-374 (1983) and Mansfield et al., Ann Allergy 55:541-543 (1985).
(4) The English usage in the manuscript is quite good but some editing would be an improvement. Several English corrections are noted:
Line 128: “criterium” should be “criterion”
Line 131: Change to read “…generally a synonym…”
Line 341: Change to read “…relevance, and an increase in the knowledge…”
Line 354-355: Change to “discovery
Good but could be improved.
Author Response
We thank very much the Reviewer for the revision of our manuscript and for all the comments and suggestions, which have been carefully considered and exploited to improve the manuscript. We hope that this revised version is now suitable for publication.
Please find below our detailed answers to the Reviewer’s comments.
Reviewer 2
Comments and Suggestions for Authors
This manuscript describes results obtained with a cohort of Italian patients regarding sensitization to cysteine protease allergens. The cysteine proteases from dust mite, Der p 1 and Der f 1, were shown to be more clinically significant than homologous cysteine proteases from several fruits – kiwi, pineapple, papaya, and fig. The manuscript was quite interesting to read and the observations were noteworthy in the sense that the investigation indicated that proteins with similar biological functions can have differing sequence identities and differing clinical significance. This manuscript could use some improvement.
COMMENT 1
(1) The authors do not describe the limitations of their study. The major limitation is that the basis for the allergy diagnosis is the history provided by the patient. The findings would be more impactful if oral challenges had been conducted to confirm the allergy or if basophil activation assays had been performed using the sera of patients.
ANSWER 1
To address the Reviewer comment, the following text has been added in the Discussion, at lines 407-410 “However, a limitation of this study is that the allergy diagnosis is based on the history provided by the patient and it was not confirmed by oral challenges and/or basophil activation assays. For this reason, a further investigation will be addressed in a near future study.”
COMMENT 2
(2) While the manuscript is interesting to read, the authors seem to repeat key observations several times within the text. The manuscript could be more succinctly presented without losing any major points.
ANSWER 2
The manuscript has been checked and some repetitions and redundancies have been deleted.
COMMENT 3
(3) Table 1 – Papain is known to be an ingestion allergens – admittedly on a rare basis. Should be included in this table. Key citations are Mansfield and Bowers, J Allergy Clin Immunol 71:371-374 (1983) and Mansfield et al., Ann Allergy 55:541-543 (1985).
ANSWER 3
The suggested details have been included in Table 1
COMMENT 4
(4) The English usage in the manuscript is quite good but some editing would be an improvement.
Several English corrections are noted:
Line 128: “criterium” should be “criterion”
Line 131: Change to read “…generally a synonym…”
Line 341: Change to read “…relevance, and an increase in the knowledge…”
Line 354-355: Change to “discovery
ANSWER 4
English has been checked and suggestions have been included